# Genotypic and Phenotypic Characterization of Erythromycin-Resistant *Staphylococcus aureus* Isolated from Bovine Mastitis and Humans in Close Contact

**DOI:** 10.3390/tropicalmed8010026

**Published:** 2022-12-29

**Authors:** Zainab Rasool, Hadiqua Noreen, Asfa Anjum, Azka Rizvi, Ali A. Rabaan, Muhammad A. Halwani, Amal A. Sabour, Mohammed Aljeldah, Basim R. Al Shammari, Salah M. Alhajri, Ibrahim H. Alshubaith, Mohammed Garout, Sehrish Firyal, Naveed Ahmed

**Affiliations:** 1Institute of Biochemistry and Biotechnology, University of Veterinary & Animal Sciences, Lahore 54000, Pakistan; 2Department of Medical Education, Avviceena Medical College, Lahore 54000, Pakistan; 3Department of Medical Education, University of Lahore, Lahore 54590, Pakistan; 4Department of Microbiology, Pakistan Kidney and Liver Institute & Research Center (PKLI & RC), Lahore 54000, Pakistan; 5Molecular Diagnostic Laboratory, Johns Hopkins Aramco Healthcare, Dhahran 31311, Saudi Arabia; 6College of Medicine, Alfaisal University, Riyadh 11533, Saudi Arabia; 7Department of Public Health and Nutrition, The University of Haripur, Haripur 22610, Pakistan; 8Department of Medical Microbiology, Faculty of Medicine, Al Baha University, Al Baha 65799, Saudi Arabia; 9Department of Botany and Microbiology, College of Science, King Saud University, Riyadh 11451, Saudi Arabia; 10Department of Clinical Laboratory Sciences, College of Applied Medical Sciences, University of Hafr Al Batin, Hafr Al Batin 39831, Saudi Arabia; 11Infectious and Zoonotic Diseases Division, Ministry of Environment, Water and Agriculture, Al-Ahsa 11116, Saudi Arabia; 12Department of International Organisations and Health Cities Al-Ahsa Municipality, Al-Ahsa 31982, Saudi Arabia; 13Department of Community Medicine and Health Care for Pilgrims, Faculty of Medicine, Umm Al-Qura University, Makkah 21955, Saudi Arabia; 14Department of Microbiology, Faculty of Life Sciences, University of Central Punjab, Lahore 54000, Pakistan; 15Department of Medical Microbiology and Parasitology, School of Medical Sciences, Universiti Sains Malaysia, Kubang Kerian 16150, Malaysia

**Keywords:** antimicrobial resistance, buffaloes, dairy animals, dairy industry, genetic diversity, zoonotic diseases

## Abstract

*Staphylococcus aureus* (*S. aureus*) is a major causative agent of mastitis and is resistant to many antibiotics. Thus, there is a need to characterize the genetic determinants of *S. aureus* erythromycin resistance, such as *ermA*, *ermB* and *ermC*. The current study aimed to determine the phenotypic and genotypic erythromycin resistance profile and relatedness of *S. aureus* recovered from bovine mastitis and humans in close contact. A total of 14 mastitis-infected buffalo milk samples and 16 samples from their respective milkers were collected from different farms of Lahore, Pakistan. The antibiotic resistance profile was determined through the disk diffusion test. The overall prevalence of *S. aureus* in mastitis-affected buffaloes was found to be 75%, of which 52.1% were resistant to erythromycin and 42.8% to clindamycin. *S. aureus* isolates recovered from milker nasal samples showed 56.25% resistance to erythromycin and 44% resistance to clindamycin. Genotypic antibiotic resistance profiles were determined from 14 milk samples through PCR. Overall, eight (52.1%), three (21.4%) and five (35.7%) *S. aureus* isolates were positive for the *ermA*, *ermB* and *ermC* genes, respectively. Moreover, 16 milker nasal *S. aureus* isolates were also tested for the presence of *ermA*, *ermB* and *ermC* genes. The *ermA*, *ermB* and *ermC* genes were observed in nine(56.7%), five (31.3%) and seven (43.7%) isolates, respectively. A significant association was shown between phenotypic and genotypic erythromycin resistance. The results indicate both that there are sufficient genetic similarities, and the actual transmission of erythromycin resistance genes between these two hosts of *S. aureus.*

## 1. Introduction

Mastitis is one of the most common emerging infections of milking animals, and affects the quantity and quality of milk. It involves the inflammation of mammary glands, the etiology of which can be infectious or non-infectious [1]. It causes irritation in the parenchyma of the mammary organs, typically caused by bacteria attacking the udder, multiplying there, and delivering harmful toxins to the mammary organs [2,3]. Mastitis has challenged economies worldwide, directly or indirectly, including Pakistan. In Pakistan, it has caused an alarming situation, and significant attention needs to be given to its control [4,5]. In dairy cows, mastitis leads to financial crises due to decreased milk production, treatment costs, including antibiotic treatments, and pre-emptive culling [2].

Of the hundreds of causes of mastitis known today, the most typical etiological agents reported are *Staphylococcus aureus* (*S. aureus*), *streptococcus agalactiae*, and others [6]. In buffalo, 70–80% cases of mastitis are due to *S. aureus* [7]. *S. aureus* is a pathogenic bacterium that is considered a threat to both animals and humans [8,9]. It causes approximately one-third of buffalo mastitis cases. In Pakistan, studies have shown that *S. aureus* is the primary infectious agent that causes mastitis [10]. For years, erythromycin has been used in treatment for various infections, and has been elected as an alternative to penicillin, cephalosporin, and other beta lactams for Gram-positive microbes [11,12]. However, resistance to erythromycin in *S. aureus* has prevailed, as its activity has resulted in the production of methylase encoded with the *erm* genes. Reports demonstrate that the genes *ermA*, erm B and erm C in bovine mastitis isolates are responsible for erythromycin resistance [13].

Antibiotic therapies are frequently ineffective in treating *S. aureus* infections, due to some of the unique features of the pathogen such as: the ability of the organism to colonize and produce micro-abscesses in the mammary gland which leads to be protected from normal defense mechanisms, the potential of invading bovine mammary epithelial cells, the switching to the small-colony variant (SCV) phenotype and biofilm formation which are relevant to chronic and recurrent infections, the capability of persisting within phagosomes, the ability to convert to L-form when exposed to antibiotics, and the ability to produce toxins [14]. Because of the 2low clearance rate in *S. aureus* associated mastitis, procedures have been developed to continue treatment for 6 to 8 days to maintain therapeutic levels of antibiotics [15]. Therefore, the overuse of antibiotics has resulted in the rise of multidrug-resistant *S. aureus*. It is well-known that excessive antimicrobial use is the main driver of antibiotic resistance [3].

*S. aureus* usually causes subclinical mastitis, which is very difficult to treat (due to high seeding rate of infected animals and chronic recurrent infections) [10]. This is probably related to the numerous virulence factors present in this pathogen, such as the ability to produce biofilm, or the production of toxins, or various enzymes designed to damage and better occupy the infected area [16]. On the other hand, it is also related to the occurrence of antibiotic resistance among these bacteria, which has arisen from the overuse of these active substances in veterinary medicine and in agriculture. Although the problem of *S. aureus* in dairy cattle has been known for years, a perfect preventive therapy or treatment has not yet been developed. This is due to the rapid genetic variability of this pathogen and the lack of knowledge about the interaction between the bacterium and the host. Studies from around the world have shown that there is no clear pattern in the distribution of virulence genes (adhesins or toxins) among bovine isolates [3]. Depending on the area from which the strains originate, a different antibiotic resistance profile can be observed. Such a situation significantly complicates the results of mastitis prevention research and the development of new therapies, and further research towards a better understanding of these bacteria should be conducted [17].

The prevalence of *S. aureus* in bovine mastitis in Pakistan is very high, and the problem is affecting the dairy industry. Many therapeutic strategies to treat mastitis have failed due to the antibiotic resistance shown by *S. aureus* [18]. Due to the high concern given to zoonotic infection, determination of the phenotypic and genotypic antimicrobial resistance profile of *S. aureus* recovered from mastitis-infected milk and the nasal carriages of farm personnel is important. Published data in antibiotic resistance-related studies on mastitis-derived *S. aureus* and on the molecular characterization and phylogenetic analysis of erythromycin resistance in *S. aureus* recovered from bovine mastitis and farm personnel in Pakistan are sparse. This study investigated strain-relatedness, the presence of genetic determinants of antibiotic resistance, and potential sources of zoonotic infections by applying phenotypic and genotypic methods. This will give an insight into the zoonotic characteristics, sources of origin, and antibiotic resistance profiles of *S. aureus* in Pakistan, as well as the control methods required.

## 2. Materials and Methods

### 2.1. Sampling Area

The current study was conducted by the Institute of Biochemistry and Biotechnology, University of Veterinary and Animal Sciences, Lahore, Pakistan from 10 August 2021 to 25 June 2022. The study was based upon buffaloes with clinical and subclinical bovine mastitis in dairy farms in Lahore. Three different locations of private dairy farms were selected for sample collection. In total, thirteen farms were included in this study. The herd size of each farm selected varied between 6 and 90 animals. Appendix A describes the sampling site, No. of total animals, and diseased animals. Lactating buffaloes with mastitis and their milk men were included in this study. A structured questionnaire was designed to collect data regarding the location of the farm, farm owner’s name, herd size, No. of animals suffering from mastitis, lactation period, antibiotics given, and No. of infected udders.

### 2.2. Screening of Infected Animals

Buffaloes with clinical and subclinical mastitis were tested in this study. Clinical mastitis was easily detected via visible symptoms such as inflammation of the udder or by impaired milk quality and appearance, while subclinical mastitis was screened via a test known as the Surf Field Mastitis Test (SFMT) [19]. In milk infected with subclinical mastitis, somatic cell count was greater than normal. The rupturing of somatic cells occurs in milk with the addition of detergent, and as a result DNA is released. Acidic DNA and basic detergent form a gel. The gel’s consistency depends on the somatic cell count.

#### 2.2.1. Collection of Mastitis Milk Samples

An aseptic collection of milk samples for the biological assay was performed. Prior to testing, nipple closes were sanitized. A total of 10 mL was collected in a sterile falcon tube via a drain test [20]. After the collection of samples, these were stored immediately at 4 °C for further processing in the lab.

#### 2.2.2. Sampling of Milker’s Nares

The sterile swabs were introduced into milkers’ nares and rotated in full rotation three times. Swabs were put back into the transport tube and labelled. These swabs were transported back to the lab (Appendix A).

### 2.3. Isolation and Identification of S. aureus from Samples

The genera-specific isolation of *Staphylococci* was performed on staph110 agar plates via the simple streaking of milk and nasal samples. Staph 110 is a selective media used to grow the genus *Staphylococcus.* The milk samples (0.1 mL) were inoculated on staph 110 agar plates using a disposable wire loop. Then, 24–48 h of incubation was completed at 37 °C. Nasal swabs were also cultured on staph 110 agar plates by rolling swabs on the plates, which were then incubated for 24–48 h at 37 °C. The purification of the representative colonies of *S. aureus* was performed on crispy mannitol salt agar (MSA) plates after isolation on staph 110 plates [21].

Specific colonies of *S. aureus* were isolated and purified by culturing distinct colonies from the *Staph* 110 plates on mannitol salt agar (Oxoid, Basingstoke, UK) by streaking the plates with a sterile inoculation loop following the 24–48 h of incubation at 37 °C. After sufficient incubation time, *S. aureus* colonies were identified by the appearance of yellow zones on the plates, indicating the fermentation of mannitol by *S. aureus*. This occurred because the byproduct produced as a result of mannitol fermentation is acidic in nature, and turns phenol red to a yellow color. Later, plates were stored at 4 °C to perform further biochemical identification tests and Gram staining [3].

#### *S. aureus* Biochemical Identification

Conventional biochemical tests were utilized for the identification of *S. aureus*. Catalase and tube coagulase was done from isolated colony of each bacterial culture and results were recorded [22].

### 2.4. Bacterial DNA Extraction

The bacterial culture was grown in nutrient broth at 37 °C for 24 h. By spinning down the bacterial culture at a high speed of 6000 rpm at 4 °C for 15 min, the palette settled in a 50-milliliter falcon tube. The supernatant was discarded and 3 mL of TEN buffer was added to the tube containing the pellet. Suspension of the pellet was performed in TEN buffer by placing it on a vortex for agitation, and then it was centrifuged at 6000 rpm at 4 °C for 15 min. Then, the supernatant was disposed of. The pellet in the tube was then treated with 2ml of freshly prepared SET buffer. Around 200–250 µL lysozyme (10 mg/mL) was added to the tube with the pellet and incubated for 30 min at 37 °C. After incubation, the pellet was treated with proteinase K, followed by the overnight incubation of samples at 50–55 °C. Upon completion of the incubation period, 500 µL of 10% SDS and 2 mL of TEN buffer were added and then incubated at 60 °C for 10 min. Equal volumes of chilled phenol: chloroform (1:1) and 1ml of 5 M NaCl were added and mixed properly. Then, centrifugation was performed for 15 min at 6000 rpm. Three visible layers were formed, and the lowest layer was phenol. The uppermost layer contained the DNA, and was then shifted to another tube. The precipitation of DNA was then carried out by the addition of chilled absolute ethanol in double volume to the aqueous layer and incubated for 30 min at −20 °C. After the designated time of incubation, tubes were centrifuged at 12,000 rpm for 15 min to obtain the pellet in intact form while the supernatant was wasted. Washing of the DNA pellet was conducted with 5ml of freshly prepared 70% ethanol. Tubes were centrifuged at 6000 rpm for 5 min. Washing of the DNA was performed three times. The pellet obtained was left to air dry overnight. Then, 300 µL of *nuc*lease-free water was added and transferred to a labeled 1.5 mL Eppendorf [23]. Spectrophotometry for the quantification of genomic DNA was performed, and the storage of samples for later use was undertaken at −20 °C.

### 2.5. Agarose Gel Electrophoresis

The quality of the extracted plasmid and whole genomic DNA was analyzed via gel electrophoresis. To make 1.2% agarose gel, 100 mL TAE 1× buffer was added to a flask and 1.2 g weighed agarose was mixed in the flask containing the buffer. After proper stirring, the solution was microwaved for 2 min until it turned transparent. Then, it was allowed to cool down to 50–60 °C, followed by the addition of 10 µL ethidium bromide. The gel caster apparatus was set with the comb in it. The agarose solution was poured in the caster and left at room temperature to polymerize. The comb was carefully taken out after the solidification of the gel. The gel caster was then placed in an electrophoresis tank filled with same TAE 1X buffer solution used to prepare the agarose gel. Samples of DNA in ratio 5:3 (5 µL of DNA sample mixed with 3 uL of 6× loading dye, bromophenol blue) were loaded in the wells of the gel. Gel electrophoresis at 110 volts was applied to the gel for 25 min [24]. After the given time, the gel was visualized under UV light to identify the quality of the DNA bands.

### 2.6. Quantification of DNA with Spectrophotometer

Quantification of the DNA sample was performed with a Nanodrop 2000 spectrophotometer. The principle of spectrophotometry relies on the basis of the absorption of the specific wavelength of ultraviolet light by *nuc*leic acid. The measurement of the absorption pattern defines the quantity of the *nuc*leic acid. Thus, all samples were quantified via spectrophotometry (Thermo Fisher Scientific, Waltham, MA, USA) and the concentrations are listed. Contamination in the *nuc*leic acid was determined via the A280/260 ratio.

### 2.7. S. aureus Molecular Identification

Whole genomic extracted samples of DNA were taken for the amplification *nuc*lease (*nuc*) gene. For the specific identification of *S. aureus*, the thermo*nuc*lease gene was used. Reported specific primers of the *nuc* gene from already published papers were taken for this purpose. Appendix A shows the list of primer sequences. The PCR reaction mixture recipe is presented, and optimized PCR conditions are listed in Appendix A. Notably, 2% agarose gel was used to visualize the results of the PCR products [25] (Appendix A).

### 2.8. Antimicrobial Susceptibility Test

Samples positive for *S. aureus* were subjected to the antibiotic susceptibility test (AST) using the disk diffusion method in order to determine their antibiotic susceptibility patterns. The results were interpreted according to Clinical and Laboratory Standards Institute (CLSI) guidelines [25,26]. Samples were simultaneously tested for cefoxitin screening by using Cefoxitin disc (30µg) and results were recorded. Antibiotic discs used were from Oxoid^TM^ and their details are as follows: Amikacin (30 µg), Chloramphenicol (30 µg), Ciprofloxacin (5 µg), Cotrimoxazole (1.25/23.75 µg), Clindamycin (2 µg), Erythromycin (15 µg), Fusidic acid (10 µg), Gentamicin (10 µg), Linezolid (30 µg), Tetracycline (30 µg), Teicoplanin (30 µg), Tigecycline (15 µg), Tobramycin (10 µg) and E strip of Vancomycin.

### 2.9. Molecular Identification of Antibiotic Resistance Genes

#### 2.9.1. Primer Design of the *ermA* Gene

A single-pair gene-specific primer set was designed using primer 3 software for the amplification of the full-length gene. The primer sequence of the *ermA* gene was developed on the basis of the plasmid-based sequence of *ermA* submitted to NCBI with accession number NC019144. The primer’s specificity was checked via NCBI BLAST.

#### 2.9.2. Primer Design of the *ermB* Gene

A single set of primers was designed to amplify the full length of the sequence of the *ermB* gene based on the chromosomal *nuc*leotide sequence of *S. aureus* submitted to NCBI with the accession number CP020741.1. The primers were designed using the primer 3 platform, and their specificity was determined via NCBI BLAST.

#### 2.9.3. Primer Design of the *ermC* Gene

Two sets of overlapping, flanking and specific primers were required to design and amplify the overlapping sequence segments of the *ermC* gene. They were designed on the basis of the plasmid *nuc*leotide sequence already published and submitted to NCBI for *S. aureus* (M17990) [13]. The primers were designed with the primer 3 platform, and their specificity was determined via NCBI BLAST. The list of primers used is shown in Appendix A.

#### 2.9.4. *ermA* Gene Amplification

For the amplification of the *ermA* gene, plasmid DNA was consumed. Only a single set of specific primers was designed for the amplification of the *ermA* gene. The PCR reaction mixture composition and the conditions optimized for PCR to amplify the gene details are given in Appendix A.

#### 2.9.5. *ermB* Gene Amplification

For the amplification of the *ermB* gene, whole-genome chromosomal DNA was taken. A single set of specific primers was designed which was used for the amplification of the *ermB* gene. The PCR reaction mixture composition and the conditions optimized for PCR to amplify the gene details are given in Appendix A.

#### 2.9.6. *ermC* Gene Amplification

For the amplification of the *ermC* gene, plasmid DNA was consumed. Two sets of specific primers were designed. One set was used first for identification, and then both sets of primers were applied for the amplification of the *ermA* gene. The PCR reaction mixture composition and the conditions optimized for PCR to amplify the gene details are given in Appendix A (Appendix A).

### 2.10. Statistical Analysis

Chi-square with the help of SPSS version 20 was used to statistically analyze the phenotypic and genotypic data.

## 3. Results

### 3.1. Genus Staphylococci Isolation on Staph 110

Of the 19 buffalo isolates and 21 nasal carriage isolates, 18 (94.7%) and 20 (95.2%) showed growth on staph 110 agar (Table 1 and Table 2). The resulting cultures showed mixed colonies. The representative colonies of Staphylococci on staph 110 agar were further purified and isolated on mannitol salt agar (MSA) for the identification of *S. aureus* (Appendix A). The identification of *S. aureus* by Gram staining and biochemical test has been shown in Appendix A).

### 3.2. Whole Bacterial Genomic and Plasmid DNA Extraction

The whole-genome bacterial DNA extracts and plasmid DNA extracts had different concentrations in the *S. aureus* isolates, the values of which ranged between 80 ng/µL and 164 ng/µL. A 280/260 value shows good ratios, ranging from 1.75 to 2.1.

### 3.3. DNA Concentration and Quality Identification via Gel Electrophoresis

Isolated whole bacterial genomic and plasmid DNA samples were run through gel electrophoresis to determine DNA concentration and quality (Appendix A).

### 3.4. Molecular Identification of S. aureus

All 31 samples that showed positive results in the microbiological identification tests were subjected to the species-specific identification of *S. aureus* with nuc gene amplification. The nuc gene was identified in all 31 isolates, and a 395 bp long amplified product was visualized after gel electrophoresis was performed. Of the 15 positive mastitis milk samples, 14 (93.3%) were positive for *S. aureus*. Significantly, 16 (100%) of the milker nasal samples were positive for *S. aureus* isolates (Table 3 and Appendix A).

### 3.5. Prevalence of S. aureus in Human Nasal and Buffalo Milk Samples

The total number of samples was 40, of which 19 were mastitis-infected buffalo milk samples and 21 samples were nasal samples from milkers directly in contact with the infected animals. According to the microbiological test and the molecular identification results, *S. aureus* was found in 14 of the 19 samples of milk from buffaloes with mastitis, and in 16 of the 21 nasal samples from farm workers (Table 4).

### 3.6. Antibiotic Resistance Profile of S. aureus

Antibiotic resistance was determined in the milk samples of 14 mastitic buffaloes. Eight (53.1%) *S. aureus* isolates were resistant to erythromycin, while six (42.7%) were sensitive. Sixteen milker nasal samples were also subjected to the antibiotic susceptibility test. Resistance to erythromycin was found in nine (56.3%) isolates, while seven (43.8%) isolates were sensitive to it (Table 5 and Appendix A).

### 3.7. Molecular Characterization Profile of Antibiotic Resistance in S. aureus

Genotypic antibiotic resistance profiles were determined in 14 milk samples from mastitic buffaloes. Eight (53.1%), three (21.4%) and five (35.7%) *S. aureus* isolates were positive for the *ermA*, *ermB* and *ermC* genes, respectively. However, the *ermA*, *ermB* and *ermC* genes were not present in 6 (42.8%), 11 (78.57) and 9 (64.2%) isolates. Sixteen milker nasal *S. aureus* isolates were also tested. *ermA*, *ermB* and *ermC* genes were observed in nine (56.3%), five (31.3%) and seven (43.6%) isolates, respectively. Notably, seven (43.7%), eleven (68.75%) and nine (56.2%) nasal samples were negative for the *ermA*, *ermB* and *ermC* genes, respectively.

### 3.8. Amplification of ermA, ermB and ermC Genes

All three resistant *ermA*, *ermB* and *ermC* genes were amplified for sequencing purposes. For the *ermA* and *ermB* genes, a single set of primers was enough to amplify the whole length of the gene, and two sets of primers were used to amplify the *ermC* gene’s full length. Samples positive for *ermA*, *ermB* and *ermC* genes in *S. aureus* isolates from the milk of mastitis-infected buffaloes and their respective milkers were selected for sequencing. (Appendix A).

## 4. Discussion

Bovine mastitis is a clinically important disease found in dairy animals. Pakistan has an abundance of small household farms where buffaloes are the major producers of milk [18]. Interaction between buffaloes and humans can lead to the transfer of pathogens. Antibiotic-resistant *S. aureus* is the most pathogenic and contagious pathogen causing bovine mastitis [4,27]. Thus, this study was conducted with the objective of evaluating the association between phenotypic and genotypic erythromycin antibiotic resistance in *S. aureus* isolated recovered from mastitis-infected buffaloes and their milkers in dairy farms of Lahore, Pakistan. Another objective was to analyze the genetic similarities between the *ermA*, *ermB* and *ermC* genes in *S. aureus* isolates from bovine mastitis cases and farm personnel. The phylogenetic analysis of antibiotic resistance genes *ermA*, *ermB* and *ermC* in *S. aureus* isolates was also performed.

In total, 30 of the 40 samples considered were positive for *S. aureus*. Of the 30 samples, 14 were milk samples and 16 were nasal samples from milkers. The overall prevalence of *S. aureus* in mastitis-affected buffaloes was found to be 75%. This prevalence is similar to that reported in a recently published study by [8]. However, this prevalence is higher than reported by Ashfaq and Muhammad in 2008 [28]. According to their study in Pakistan, *S. aureus* was found to be positive in 48.08% of isolates from dairy buffaloes.

The resistance of *S. aureus* to antimicrobial agents is an increasing global concern. The determination of antimicrobial resistance profiles is necessary to develop effective therapeutic strategies [29]. The monitoring of resistant strain spread should also be considered important. In this study, the antimicrobial susceptibility profiles of *S. aureus* were determined and high levels of resistance to erythromycin (52.1%) followed by clindamycin (42.8%) were detected. This resistance rate is comparable to one found by another study Mahdavi and colleagues in which they reported 48% and 46% resistance to erythromycin and clindamycin [30].

Results of our study are similar to the study conducted in China, where they concluded that most of the common drugs such as Erythromycin, Tetracycline, Clindamycin, penicillin and cotrimoxazole are ineffective against *S. aureus* isolated from bovine mastitis samples. Our study also showed higher resistance of *S. aureus* against these drugs [31]. Antibiotic resistance was determined in 14 milk samples from mastitic buffaloes, and 47% of *S. aureus* isolates were resistant to erythromycin. Clindamycin resistance was shown in 33% of isolates. It was also found that genotypic and phenotypic erythromycin resistance are associated. The genotypic antibiotic resistance profiles of 14 milk samples from mastitic buffaloes were determined. Overall, 52.1%, 21.4% and 35.7% of *S. aureus* isolates were positive for the *ermA*, *ermB* and *ermC* genes, respectively. Sixteen milker nasal *S. aureus* isolates were also tested for the presence of *ermA*, *B* and *C* genes, and these genes were observed in 56.2%, 31.3% and 43.8% of the samples, respectively. This finding is in correspondence to another study conducted in China, according to which highest number of genes detected from *S. aureus* was *ermA* gene [32].

In small household farms where there is interaction between animals and humans, there is suitable opportunity for microbial organisms to colonize humans [33]. However, some reports have noted that *S. aureus* is host-specific, but this applies to some specific lineages [16,26]. Most of them do not show host-specificity. Apart from these characteristics, *S. aureus* is an adaptive pathogen, and can adapt to many unfavorable circumstances [14]. Some genetic adaptions and mutations have helped it to become a successful pathogen [34].

To investigate the transmission of antibiotic resistance genes between buffaloes and humans via *S. aureus* transfer, the *ermA*, *B* and *C* genes were analyzed. This showed that there is little or no difference between *S. aureus* isolates of buffaloes and humans who live in close contact. Another study conducted on erythromycin resistant *S. aureus* isolated from environmental samples also reported the comparable results in sensitivity pattern of *S. aureus*, indicating that origin of isolates is independent of the sensitivity pattern [35]. These bugs can be acquired from environment by both milkers and buffaloes, which ultimately leads to infection [16]. This study reports that there is a transmission of antibiotic-resistant *S. aureus* between buffaloes and closely related humans, as the *ermA*, *ermB* and *ermC* gene sequences from buffaloes and their milkers from same farm were closely related, but distantly related to those isolated from other farms.

Study Limitations: The current study did not report the data on clinical and subclinical mastitis because of the restriction in ethical approval from the target institutions. Furthermore, because of the financial issues, the sequencing analysis of PCR products could not be done in order to further identify the isolates as well as to check their phylogenetics.

## 5. Conclusions

The current study concludes that erythromycin-resistant *S. aureus* was prevalent among buffaloes with mastitis and farm workers in dairy settings of Lahore, Pakistan. Hence, it is proven that there is an association between phenotypic and genotypic erythromycin antibiotic resistance. Antibiotic resistance genes (*ermA*, *B* and *C*) of *S. aureus* isolates from buffaloes with mastitis are genetically related to those found in their farm workers’ isolates. Therefore, there is a possibility of zoonotic transfer.

## Figures and Tables

**Table 1 tropicalmed-08-00026-t001:** Microbiological identification test results of mastitic buffalo *S. aureus* isolates.

Sr. No.	Identification Test	No. of Total Samples	No. of Positive Samples	No. of Negative Samples	Positive Samples
1.	Growth on staph 110	19	18	1	94.7%
2.	Mannitol salt agar	16	15	1	93.7%
3.	Gram staining	15	15	0	100%
4.	Catalase test	15	15	0	100%
5.	Tube coagulase	15	12	3	80%

**Table 2 tropicalmed-08-00026-t002:** Microbiological identification test results of milkers’ nasal *S. aureus* isolates.

Sr. No.	Identification Test	No. of Total Samples	No. of Positive Samples	No. of Negative Samples	Positive Samples
1.	Growth on staph 110	21	17	1	80.95%
2.	Mannitol salt agar	17	16	1	94.11%
3.	Gram staining	16	16	0	100%
4.	Catalase test	16	16	0	100%
5.	Tube coagulase	16	12	4	75

**Table 3 tropicalmed-08-00026-t003:** Results of molecular identification of *S. aureus* isolates from buffalo and milkers.

Sr. No.	Origin of Isolates	Total No. of Samples	Total No. of Positive Samples	Total No. of Negative Samples	Positive Samples
1.	Buffalo	15	14	1	93.3%
2.	Milker	16	16	0	100%%

**Table 4 tropicalmed-08-00026-t004:** Prevalence of *S. aureus* in mastitic milk and milker nasal samples.

Sr. No.	Characteristics	Buffalo Mastitic Milk Samples		Milker Nasal Samples	
		*n*	%	*n*	%
1.	Total No. of samples	19	100%	21	100%
2.	Prevalence of *S. aureus*	14	73.6%	16	76.19%

**Table 5 tropicalmed-08-00026-t005:** Antibiotic resistance profile of Staphylococcus isolated from mastitis buffalo milk samples and milker’s nasal swab samples.

Antibiotics	Resistance % Milker Nasal Samples	Resistance % Mastitis Buffalo Milk Samples
Amikacin	0	0
Clindamycin	45.5	47.2
Erythromycin	56.3	53.1
Gentamycin	43.0	39.4
Tobramycin	35.8	37.5
Teicoplanin	0	0
Tigecycline	16.8	18.0
Tetracycline	62.3	61.4
Vancomycin	0	0
Linezolid	0	0
Ciprofloxacin	68.7	70.5
Cotrimoxazole	39.9	37.6

## Data Availability

More data related to this study could be available upon a reasonable request to the corresponding authors.

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
