# Peer review of "Genotypic and Phenotypic Characterization of Erythromycin-Resistant Staphylococcus aureus Isolated from Bovine Mastitis and Humans in Close Contact"

_tropicalmed, 2022, doi:10.3390/tropicalmed8010026_

Round 1

Reviewer 1 Report

The authors declared, “Genotypic and phenotypic characterization of erythromycin resistant Staphylococcus aureus isolates of zoonotic origin”. Despite the importance of the study, the article lacks a good presentation. It has many grammar and language mistakes.

The order of event writing should be the same either in abstract, introduction, material, ……….so on.

The title is not informative and must be improved.

An expert in the English language should revise it before publishing. The following major points must be taken into consideration:

Abstract:

-Line 41  remove out

- Provide in the abstract an informative and balanced summary of what was done and what was found.

-Arrange the keywords in alphabetic order

Introduction:

-The introduction needs to be more informative. The introduction should be improved(illustrating the aim of the work and some missing data about the public health importance concerning the emergence of multidrug-resistant (MDR) bacterial pathogens that reflect the necessity of new potent and safe antimicrobial agents. Several studies proved the widespread MDR- bacterial pathogens):

-Give a hint about the virulence factors and the mechanism of disease occurrence, and infections caused by Staph aureus.

-Authors mention MRSA and there is no work on MRSA so authors should remove it from the introduction

·        Line 63 serious changes to a serious

·        Line 67,are not in practice change to is not in practice

·        Line 84 alternative changes to an alternative

·        Line 88 responsible to changes to responsible for

·        Line 108 have been failed changes to have failed

Methods

-Line 125 13 change to thirteen

- Line 133 easily be detected chang to easily detected

- Line 136 occur change to occurs

-Line 145 rotated change to and rotated

-Line 151 please remove platenium

- Line 151,152  please rephrase this sentence “0.1 ml quarter of foremilk sample from each tube on staph 110 plates of agar streaking was done”

- please add a reference of isolation and identifictaion of S. aureus from samples

- please add manufacturer of different media and antibiotics

- Line 203 samples were changed to samples was

-There are many references are missed

-There are many grammatical editing must be improved

-Line 251 were use change to were used

- The authors performed antibiotic susceptibility tests on S. aureus isolated, without mentioning the concentration of used antibiotics . Please provide details on the data for establishing the antibiotic susceptibility patterns.

- please add statistical methods and program used in this manuscript

-please add other biochemical tests to confirm isolation of Staph 

Results

- Authors do not mention anything on prevelance of mastitis please provide the prevalence of clinical and sub-clinical mastitis

-  Line 261 please remove the manufacturer company and add in the materials section

-Line 264 mix changed to mixed

-Please add a table of antimicrobial results

Discussion and conclusion

-It is the opinion of the reviewer that the discussion section needs to be reviewed.  Authors need to improve discussion and explore the significance of the study compared to other studies.

- Give a cautious overall interpretation of results considering objectives, limitations, the multiplicity of analyses, results from similar studies, and other relevant evidence.

-Line 388 helps change to help

- it is better to report the consistency and inconsistency of the present research with other relevant evidence, regarding each antibiogram and MDR.

 -For correct identification of the association between phenotypic and genotypic erythromycin antibiotic resistance ,DNA sequencing should be applied for the PCR products 

Authors should go through the manuscript very carefully and correct these mistakes so that this work reads scientifically more correct                                                                                             

Author Response

Reviewer 1

Comments and Suggestions for Authors

The authors declared, “Genotypic and phenotypic characterization of erythromycin resistant Staphylococcus aureus isolates of zoonotic origin”. Despite the importance of the study, the article lacks a good presentation. It has many grammar and language mistakes.

Response: Dear reviewer, we would like to appreciate your efforts in reviewing our manuscript. We would also like to acknowledge that the comments from your side and other reviewers have significantly improved the quality of manuscript. We have revised the manuscript according to the suggestion from your side. We have added some information in the introduction and discussion section to make it better for the reader. We also removed all of the extra and duplicate information from the manuscript. The representation of results has been improved. Furthermore, the manuscript has been thoroughly revised for English proofreading and grammatical mistakes by the English language expert.

The order of event writing should be the same either in abstract, introduction, material, ……….so on.

Response: Dear reviewer, thank you for your valuable suggestion. We have revised the abstract in the revised version of manuscript.

The title is not informative and must be improved.

Response: Dear reviewer, we have revised the title in the revised version of manuscript.

An expert in the English language should revise it before publishing. The following major points must be taken into consideration:

Response: Dear reviewer, thank you for your valuable suggestion. The manuscript has been thoroughly revised for English proofreading and grammatical mistakes by the English language expert.

Abstract:

-Line 41  remove out

Response: Line 41: The word “out” has been removed from the revised version of manuscript.

- Provide in the abstract an informative and balanced summary of what was done and what was found.

Response: The abstract has been revised. Some sentences has been removed in order to make it clarify for the reader.

-Arrange the keywords in alphabetic order

Response: Line 54-55: Corrected.

Introduction:

-The introduction needs to be more informative. The introduction should be improved (illustrating the aim of the work and some missing data about the public health importance concerning the emergence of multidrug-resistant (MDR) bacterial pathogens that reflect the necessity of new potent and safe antimicrobial agents. Several studies proved the widespread MDR- bacterial pathogens):

Response: Line 83-109: New information has been added in the revised Introduction section. Furthermore, some paragraphs have been removed.

-Give a hint about the virulence factors and the mechanism of disease occurrence, and infections caused by Staph aureus.

Response: Line 83-109:  A hint about the virulence mechanism has been written in revised manuscript.

-Authors mention MRSA and there is no work on MRSA so authors should remove it from the introduction

Response: The respective information has been removed from the revised manuscript. Previously, this information was added because we processed our samples for cefoxitin screen as well when antibiotic susceptibility was tested.

  • Line 63 serious changes to a serious

Response: Line 62-63: The sentence has been rephrased.

  • Line 67,are not in practice change to is not in practice

Response: Line 67: Corrected.

  • Line 84 alternative changes to an alternative

Response: Line 77: Corrected.

  • Line 88 responsible to changes to responsible for

Response: Line 82: Corrected.

  • Line 108 have been failed changes to have failed

Response: Line 112: Corrected.

Methods

-Line 125 13 change to thirteen

Response: Line 131: Corrected.

- Line 133 easily be detected chang to easily detected

Response: Line 140: Corrected.

- Line 136 occur change to occurs

Response: Line 144: Corrected.

-Line 145 rotated change to and rotated

Response: Line 153: Corrected.

-Line 151 please remove platenium

Response: Line 160: The word “platinum” has been removed.

- Line 151,152 please rephrase this sentence “0.1 ml quarter of foremilk sample from each tube on staph 110 plates of agar streaking was done”

Response: Line 159-160: The sentence has been corrected.

- please add a reference of isolation and identifictaion of S. aureus from samples

Response: Line 164: The reference has been cited.

- please add manufacturer of different media and antibiotics

Response: Line 166, 240: The manufacturer has been written.

- Line 203 samples were changed to samples was

Response: Line 220: Corrected.

-There are many references are missed

Response: References has been cited in the methodology section and wherever needed.

-There are many grammatical editing must be improved

Response: Dear reviewer, thank you for your valuable suggestion. The manuscript has been thoroughly revised for English proofreading and grammatical mistakes by the English language expert.

-Line 251 were use change to were used

Response: Line 265: The sentence has been revised.

- The authors performed antibiotic susceptibility tests on S. aureus isolated, without mentioning the concentration of used antibiotics. Please provide details on the data for establishing the antibiotic susceptibility patterns.

Response: Line 238-243, Table 5: The respective information has been added.

- please add statistical methods and program used in this manuscript

Response: Line 280-281: Statistical methods applied and the software used have been added to the manuscript.

-please add other biochemical tests to confirm isolation of Staph 

Response: Line 173-176, Table 1 and 2: New information has been added.

Results

- Authors do not mention anything on prevalence of mastitis please provide the prevalence of clinical and sub-clinical mastitis

Response: Dear reviewer, thank you for your valuable suggestion. Because of ethical approval limitation we could not provide data on clinical and subclinical mastitis cases. This was the limitation of our study. I hope that you will understand the ethical issue and will allow us to proceed further. The study limitations have been added at the end of discussion section.

-  Line 261 please remove the manufacturer company and add in the materials section

Response: Line 284-289: The manufacturer information has been removed.

-Line 264 mix changed to mixed

Response: Line 286: Corrected.

-Please add a table of antimicrobial results.

Response: Table 5 has been added.

Discussion and conclusion

-It is the opinion of the reviewer that the discussion section needs to be reviewed.  Authors need to improve discussion and explore the significance of the study compared to other studies.

Response: Line 369-375, 384-386, 397-401: New information has been added. Furthermore, some of the extra information has been removed.

- Give a cautious overall interpretation of results considering objectives, limitations, the multiplicity of analyses, results from similar studies, and other relevant evidence.

Response: Line 369-375, 384-386, 397-401: New information has been provided.

-Line 388 helps change to help

Response: Line 387: Corrected.

- it is better to report the consistency and inconsistency of the present research with other relevant evidence, regarding each antibiogram and MDR.

Response: New reference studies has been added as suggested.

 -For correct identification of the association between phenotypic and genotypic erythromycin antibiotic resistance, DNA sequencing should be applied for the PCR products 

Response: Dear reviewer, thank you for your valuable suggestion. Actually, that’s a very good suggestion but since this was a self-funded project with limited resources, we could not include sequencing in this study. However, we’ll try to implement your suggestion in our next study. Furthermore, at line 406-409, the study limitations has been added.

Authors should go through the manuscript very carefully and correct these mistakes so that this work reads scientifically more correct.

Response: Dear reviewer, we would like to appreciate your efforts in reviewing our manuscript. We would also like to acknowledge that the comments from your side and other reviewers has significantly improved the quality of manuscript. 

Reviewer 2 Report

It is a good job, however it requires some adaptations in relation to writing, considering not to constantly repeat information in the content of the work.

Information is repeated in the results and discussion sections, it is suggested to be concise, in addition to not using information that seems more like a literature review in the discussion, and preferably discussing with findings from similar works.

Some details were detected that were marked in the writing as a way of correcting, both in style and in substance on the subject.

Author Response

Reviewer 2

Comments and Suggestions for Authors

It is a good job, however it requires some adaptations in relation to writing, considering not to constantly repeat information in the content of the work. Information is repeated in the results and discussion sections, it is suggested to be concise, in addition to not using information that seems more like a literature review in the discussion, and preferably discussing with findings from similar works. Some details were detected that were marked in the writing as a way of correcting, both in style and in substance on the subject.

Response: Dear reviewer, we would like to appreciate your efforts in reviewing our manuscript. We would also like to acknowledge that the comments from your side and other reviewers have significantly improved the quality of manuscript. We have revised the manuscript according to the suggestion from your side. We have added some information in the introduction and discussion section to make it better for the reader. We also removed all of the extra and duplicate information from the manuscript. Furthermore, the manuscript has been thoroughly revised for English proofreading and grammatical mistakes by the English language expert.

Reviewer 3 Report

The aim of this study is appropriate and significant for public health. However, the study described in this manuscript is too primitive and adds no scientifically novel finding. Concerns of this manuscript are listed as follows.

1. In Materials and Methods section, period and time of sample collection was not written. In such case, the presented data has no meaning of epidemiology.

2.  Numbers of isolates from buffalo milk samples and milkers were too low to represent data as epidemiological information.

3. Results section contained too much volume of conventional bacteriological contents (Gram staining, and others; 3.1, 3.2).

4. Table 1 is not necessary. It is enough to cite only Table 1. Figure 1 and 2 are also not necessary. They are not new findings.

5. Overall, presented data has no epidemiological sgnificance.

Author Response

Reviewer 3

Comments and Suggestions for Authors

The aim of this study is appropriate and significant for public health. However, the study described in this manuscript is too primitive and adds no scientifically novel finding. Concerns of this manuscript are listed as follows.

Response: Most of the past work that has been done was in other population/countries, this study was included participants from one of the biggest city of our country which included sample of zoonotic origin. Since our country is under develop this kind of information about zoonotic transfer of staphylococcus is not well documented. We do agree it might not be beneficial for public health at large but we do believe it will give adequate information required to control and evaluate zoonotic transfer of Erythromycin resistant Staphylococcus aureus in our country. Currently, National Institute for Health is working on National level to document prevalence and resistivity pattern of these bugs, so we think it can contribute to the greater cause.

  1. In Materials and Methods section, period and time of sample collection was not written. In such case, the presented data has no meaning of epidemiology.

Response: Line 128-129: Time duration has been added.

  1. Numbers of isolates from buffalo milk samples and milkers were too low to represent data as epidemiological information.

Response: Dear reviewer, thank you for the very valuable point. Thank you for different perspective and we agree with you. So we have addressed this part of manuscript as well and rephrased it (removed epidemiological aspect).

  1. Results section contained too much volume of conventional bacteriological contents (Gram staining, and others; 3.1, 3.2).

Response: The respective information has been removed from the manuscript. Since it was a self-funded study with limited resources, we have included the most convenient techniques.

  1. Table 1 is not necessary. It is enough to cite only Table 1. Figure 1 and 2 are also not necessary. They are not new findings.

Response: The table 1 and figure 1, 2 has been removed from the main manuscript document and has been shifted to the supplementary material.

  1. Overall, presented data has no epidemiological significance.

Response: Thank you for valuable input, we have addressed this part of the manuscript. Thank you for different perspective and we agree with you. We have removed the epidemiological part from the revised version of manuscript.

Round 2

Reviewer 1 Report

Accept this manuscript after improving this manuscript by authors

Reviewer 3 Report

None.